# Metacognitive Rating Scale: A Study Applying a Korean Translation to Individuals with Schizophrenia

**DOI:** 10.3390/ijerph18136853

**Published:** 2021-06-25

**Authors:** Mihwa Han, Kyunghee Lee, Mijung Kim, Youngjin Heo, Hyunseok Choi

**Affiliations:** 1Department of Nursing Science, Sunlin University, Pohang 37560, Korea; mihwahanrn@naver.com; 2College of Nursing, Keimyung University, Daegu 42601, Korea; 3College of Nursing, Masan University, Masan 51217, Korea; 01dkwnaak@hanmail.net; 4Industry-Academic Cooperation Foundation, Keimyung University, Daegu 42601, Korea; hoho321@daum.net; 5Center for Educational Performance, Keimyung University, Daegu 42601, Korea; chsuk1@kmu.ac.kr

**Keywords:** metacognition, metacognitive belief, anger expression, schizophrenia, factor analysis

## Abstract

Metacognition is a higher-level cognition of identifying one’s own mental status, beliefs, and intentions. This research comprised a survey of 184 people with schizophrenia to verify the reliability of the metacognitive rating scale (MCRS) with the revised and supplemented metacognitions questionnaire (MCQ) to measure the dysfunctional metacognitive beliefs of people with schizophrenia by adding the concepts of anger and anxiety. This study analyzed the data using principal component analysis and the varimax method for exploratory factor analysis. To examine the reliability of the extracted factors, Cronbach’s α was used. According to the results, reliability was ensured for five factors: positive beliefs about worry, negative beliefs about uncontrollability and danger of worry, cognitive confidence, need for control, and cognitive self-consciousness. The negative beliefs about uncontrollability and danger of worry and the need for control on anger expression, which were both added in this research, exhibited the highest correlation (r = 0.727). The results suggest that the MCRS is a reliable tool to measure the metacognition of people with schizophrenia.

## 1. Introduction

Schizophrenia is a serious mental illness, for which hallucinations and delusions are key symptoms [1]. Schizophrenia is characterized by delusions with fixed false beliefs due to hallucinations and strong convictions; people with schizophrenia have difficulties discerning between events occurring in the external world and those occurring inside the patient’s mind. As a result, symptom management is difficult, and social functioning, such as the ability to maintain a job, declines [2,3,4]. The cognitive bias causing, maintaining, or deteriorating the delusion symptom of schizophrenia is revealed as a selective disorder by the distortion of thoughts and handling such as memory accuracy and attention deficit rather than by the lack and limitation of mental ability [5,6].

Metacognition is a higher-order thinking process involving active control over oneself, allowing observation of one’s flow of consciousness and recalling memories. While cognition is defined as an intelligent activity to handle a task, metacognition performs monitoring of cognitive activities [7]. Namely, metacognition means the psychological structure, knowledge, events, and processes related to the control, revision, and interpretation of the thought itself [8].

Metacognition plays a role in helping people with schizophrenia identify that their thoughts, emotions, and intentions regarding specific events are biased and foster an ability to integrate them. It calls attention to their own cognitive bias when they are exposed to a threatening or anxiety-causing situation [9,10,11]. The ability of people with schizophrenia to recognize cognitive bias that distorts information collection and cognition is called metacognitive beliefs [12]. Metacognitive beliefs can predict mental illness symptoms’ frequency and effects, and dysfunctional metacognitive beliefs may cause more serious and negative effects [13,14].

In psychopathology, the self-regulatory executive function (S-REF) model, which explains metacognition, recognizes that dysfunctional metacognitive beliefs occur based on the people’s type of response to negative thoughts and emotions, namely cognitive attentional syndrome (CAS) [15,16]. According to the S-REF model, dysfunctional metacognitive beliefs on worry and intrusive thoughts are related to the development and retention of hallucinations and delusions [17]. Above all, dysfunctional metacognitive beliefs are highly correlated with anger due to a belief that one’s own thoughts cannot be controlled [18]. This does not indicate that better self-reflection was present in people with high metacognitive abilities.

To evaluate dysfunctional metacognitive beliefs using the S-REF model, the metacognitions questionnaire was originally developed with 65 items (MCQ) [19]; it was later shortened to 30 items (Metacognitions Questionnaire-30: MCQ-30) [8]. Similar to the original MCQ, the MCQ-30 consists of the following five factors that are distinguished conceptually despite their correlations: cognitive confidence (CC), positive beliefs about worry (POS), cognitive self-consciousness (CSC), negative beliefs about uncontrollability and danger of worry (NEG), and need for control (NC) [20]. The MCQ-30 consists of six items for each of the five factors and presents simple multidimensionally measured values on metacognition; thus, it is more economical and efficient than the original MCQ [8].

The research on the development of the MCQ [19] explained that thought control can produce adverse effects and make people reveal aggressive impulses; further, thought control is correlated with worry and anxiety vulnerability in the cognitive process [8,21]. Moeller [22] also reported that the five factors of the MCQ-30 are correlated and that anger is caused at the metacognitive level according to the research developing the metacognitive anger processing (MAP) scale, which explored the metacognitive factors of anger. Caselli et al. [23] observed the effects of metacognitive beliefs, rumination, and anger and found that metacognitive beliefs directly affect anger, separately from rumination. The risk of violently expressing anger in men and women with schizophrenia is 4.6 times and 23.3 times higher, respectively, compared to normal men and women [24]. Furthermore, Ringer and Lysaker [25] insist that the failure of people with schizophrenia to control anger expression reduces treatment results and responses. Despite previous research results, most studies on metacognition and anger have been conducted on college students and hospitalized patients with mixed clinical diagnoses.

To advance the existing literature, this research examined the correlations between metacognitive beliefs and anger in people with schizophrenia. Specifically, this study analyzed the reliability of the metacognitive rating scale (MCRS), which is a revised version of the MCQ-30 that is supplemented with anger and anxiety concepts, for people with schizophrenia. The specific purposes of this study are as follows:

Firstly, to verify the reliability of the MCRS’s factors.

Secondly, to verify the mean differences according to the general characteristics of the MCRS.

Thirdly, to analyze the correlations of the MCRS’s factors.

## 2. Materials and Methods

This research targeted people with schizophrenia and verified the reliability of the MCRS by adding anger and anxiety concepts, revising and supplementing the MCQ, which was developed to measure dysfunctional metacognitive beliefs in the general population (Appendix A).

The MCRS has 30 items, with six items for each of the five factors of the MCQ developed by Cartwright-Hatton and Wells [19]: positive beliefs about worry (POS), negative beliefs about uncontrollability and danger of worry (NEG), cognitive confidence (CC), need for control (NC) (anger expression), and cognitive self-consciousness (CSC). The MCRS requires respondents to rate each item on a 5-point Likert scale, ranging from one point (strongly disagree) to five points (strongly agree).

Cartwright-Hatton approved the use of the MCQ for revision and supplementation in this research to verify its reliability. The MCQ was first translated into Korean. The researchers then reviewed whether further revision was necessary in terms of the translation’s accuracy and adequacy of expression. Next, the researchers ensured that the translation did not change the meaning of each item. Finally, the researchers added anger and anxiety concepts reflecting Korean cultural features in consideration of cultural differences [26].

The data were collected after obtaining approval from the Korea University’s Institutional Review Board (IRB No. 40525-202011-HR-068-03). The study’s respondents were people with schizophrenia who were hospitalized in mental hospitals and agreed to voluntarily participate in the research after understanding the research purpose. Out of the 200 distributed questionnaires, 184 questionnaire responses were analyzed after excluding incomplete responses. To analyze the collected data, frequency analysis, descriptive statistics, factor analysis, independent samples *t*-test, and variance analysis were conducted using SPSS Statistics 26.0. (IBM Corp., Armonk, NY, USA)

## 3. Results

### 3.1. Demographic Characteristics

In terms of demographic characteristics, 59.2% of the respondents were men and 40.8% were women. As for the level of education, 53.8% of the respondents dropped out of middle school or earlier and 46.2% graduated from middle school or received a higher level of education; thus, the level of education was low overall. Concerning the marital status, 58.7% of the respondents were unmarried, 15.2% were divorced, and 26.1% selected “other.” Regarding the age, 44.6% of the respondents were 49 or under, 55.4% were 50 or over, and the mean age was 50.7 years. As for the residential type, 29.9% of the respondents lived alone, 35.9% lived with their family and friends, and 34.2% selected “other” (Table 1).

### 3.2. Analysis of Reliability

To analyze the MCRS’s reliability and validity, exploratory factor analysis (EFA) was used to infer the factors (principal components) that could potentially account for the observed correlation. To simplify the related variables, principal component analysis was used as a factor extraction method to classify each factor’s characteristics, and the method used to determine the number of factors was when the eigenvalue was 1 or higher. For the factor rotation method, the varimax method was used. To review the validity of the variables included in the extracted factors, Cronbach’s α was used to analyze reliability.

If the eigenvalue was 1 or higher; if Cronbach’s α was 0.6 or higher, reliability was confirmed [27] (Table 2).

For POS (positive beliefs about worry), reliability was ensured as Cronbach’s α was 0.643. For NEG (negative beliefs about uncontrollability and danger of worry), reliability was ensured; Cronbach’s α was 0.821, the eigenvalue was greater than 1. For CC (cognitive confidence), reliability was ensured as Cronbach’s α was 0.821. For NC (need for control) (anger expression), reliability was ensured as Cronbach’s α was 0.696. For CSC (cognitive self-consciousness), reliability was ensured as Cronbach’s α was 0.731. In sum, the reliability of the five factors was ensured.

### 3.3. Test for Differences in Means

#### 3.3.1. Test for Differences between Means by Gender

There were no significant differences between the means of POS, NEG, CC, NC, and CSC by gender. Although POS, NEG, and CSC did not show significant differences, female respondents had relatively higher mean values than male respondents. CC and NC factors also did not exhibit significant differences; however, the male respondents had relatively higher mean values than the female respondents (Table 3).

#### 3.3.2. Test for Differences between the Means by Level of Education

For *t* POS and NEC, the mean values were significantly different between the respondents who graduated from middle school graduates or received a higher level of education and the respondents who dropped out of middle school or earlier. Although no significant differences were shown for the mean values of CC, NC, and CSC by the level of education, the respondents who graduated from middle school graduates or received a higher level of education had relatively higher mean values than the respondents who dropped out of middle school or earlier (Table 3).

#### 3.3.3. Test for Differences between the Means by Age

There were no significant differences between the means of POS, NEG, CC, NC, and CSC by age. However, the respondents who were 50 or over had relatively higher mean values of POS than the respondents who were 49 or under. The respondents who were 49 or under had relatively higher mean values on NEG, NC, and CSC than those who were 50 or over (Table 3).

#### 3.3.4. Test for Differences between the Means by Marital Status

There were no significant differences in the means of POS, NEG, CC, NC, and CSC by marital status. However, the mean values for all the factors were highest for the respondents who were divorced, lower for the respondents who were unmarried, and even lower for the respondents who selected “other” (Table 4).

#### 3.3.5. Test for Differences between the Means by Residential Type

There were no significant differences between the means of POS, NEG, NC, and CSC by residential type (Table 4).

### 3.4. Correlation Analysis

As a result of analyzing the correlations between the five factors, NEG and NC (anger expression) exhibited the highest correlation, with *r* = 0.727. CC and CSC exhibited the lowest correlation, with *r* = 0.397 (Table 5).

## 4. Discussion

Metacognition refers to the ability to see oneself objectively and control oneself [7]. The main symptoms of schizophrenia are delusions and hallucinations, and proper evaluation of dysfunctional metacognition is needed to ease symptoms and improve functions of people with schizophrenia. Anger expression is highly correlated with treatment results and responses to schizophrenia [25]. In this study of people with schizophrenia, we analyzed the reliability of the MCRS, which adds anger and anxiety concepts to the MCQ [19], a questionnaire designed to evaluate personal differences in assessing worry, invasive thoughts, and beliefs on cognitive function (Appendix A).

Exploratory factor analysis was performed by classifying five factors in the same way as the existing MCQ. The reliability of the items “Worrying helps me to avoid problems in the future” and “I need to worry in order to work well” was lower than that of the items “Worrying helps me to avoid problems in the future,” “I need to worry in order to remain organized,” “Worrying helps me to get things sorted out in my mind,” and “Worrying helps me cope.” The positive beliefs on worry reflected the thought that worrying helps solve problems or develop a plan to cope. When conducting exploratory factory analysis, the reliability of the item “Worrying helps me to solve problems” was 0.459. Therefore, inconsistent responses seemed to be elicited from respondents when worry was explained as a pessimistic thought. The reliability of the item “I need to worry in order to work well” was 0.370, thereby confirming that differences existed in the respondents’ responses when worry was presented as carefully coping with a dangerous situation without mentioning worry.

Factor two—NEG—was reliable for all the six items: “My worrying is dangerous for me,” “When you become addicted to worry, even the slightest amount of stress lead to anxiety,” “My worrying thoughts persist no matter how I try to stop them,” “I cannot ignore my worrying thoughts,” “My worrying could make me go mad,” and “When I start worrying, I cannot stop.” The items represent the negative belief that worrying cannot be controlled and is dangerous. As each item’s reliability was 0.743 or higher when conducting exploratory factor analysis, these items were reliable for measuring negative beliefs on worrying for people with schizophrenia.

Factor three—CC—was reliable for the following six items: “I have little confidence in my memory for words and names,” “My memory can mislead me at times,” “I have a poor memory,” “I have little confidence in my memory for places,” “I do not trust my memory,” and “I have little confidence in my memory for actions.” The lack of cognitive confidence indicates a lack of confidence in memory and concentration. As each item’s reliability was 0.684 or higher when conducting exploratory factor analysis, these items were reliable for measuring the cognitive confidence of people with schizophrenia. However, this means there was a limitation in measuring cognitive confidence limited to personal subjective confidence in memory.

For factor four—NC (anger expression)—the items “Expressing anger is bad” and “I can express anger in words that are not aggressive” exhibited lower reliability than the following items: “I easily get angry when I feel irritated,” “I always try to suppress my anger,” “It is my weakness that I cannot control my anger,” and “I cannot control expressions of anger.” NC reflects negative beliefs including wrong beliefs, punishment, or responsibility. In this research, NC was revised and supplemented with items measuring subjective recognition of anger expression. The reliability of the factor was ensured at 0.696. In particular, the reliability of the item “If I could not control my thoughts, I would not be able to function” was 0.278, indicating that anger could be controlled differently what was stated in other items; as a result, the respondents were considered to have given inconsistent responses. Thus, there is a need to check the items related to anger expression on the NC factor through repeated research.

For factor five—CSC—the reliability of the following six items was ensured: “I think a lot about my thoughts,” “I am aware of the way my mind works when I think through a problem,” “I monitor my thoughts,” “I am constantly aware of my thoughts,” “I pay close attention to the way my mind works,” and “I constantly examine my thoughts.” CSC encompasses self-recognition that one cannot control worrying, including a thinking process committed to one’s own thoughts. In this research, the reliability of the CSC factor was ensured at 0.731. Specifically, the reliability of the item “I am aware of the way my mind works when I think through a problem” was 0.439, indicating that the negative meaning of CSC on worrying or repeatedly occurring thoughts was slightly excluded unlike the other items. Therefore, the responses of the subjects seem to show inconsistencies.

In terms of general characteristics, the mean metacognition by gender, age, marital status, and residential type did not exhibit significant differences. However, the mean values for POS (*p* = 0.009) and NEG (*p* = 0.041) did exhibit significant differences by level of education: the respondents who graduated from middle school or received a higher level of education had significantly higher mean values than the respondents who dropped out of middle school or earlier. Given that the respondents’ overall level of education was low and their mean age was 50, it would be impractical to conclude that a higher education level is related to higher metacognitive beliefs based on graduation from middle school. Nevertheless, a previous study [28] reported that self-handicapping behaviors negatively affecting academic accomplishment decrease as metacognitive beliefs increase. This explains that continuing learning, without abandoning a course of learning, is correlated with metacognitive beliefs.

The respondents’ metacognition scores for each factor (range of 14–17 points) were generally higher than those (range of 8–12 points) found in a study by Wells and Cartwright-Hatton [8], which highlights the dysfunctional metacognitive beliefs of people with schizophrenia. However, the MCRS needs to be carefully compared considering that the original tools were revised and supplemented considering cultural differences. According to a study by Østefjells et al. [29], the people with schizophrenia had higher metacognition scores for dysfunctional metacognitive beliefs than the healthy control group in almost all the factors. Further, in another study, the people with mental illness symptoms had higher dysfunctional metacognitive beliefs than the healthy people [29]. Given these findings, determining the degree of dysfunctional metacognition is useful for understanding people with schizophrenia [30].

Further, the correlation coefficient between NEG and NC (anger expression) was 0.727, which was the highest correlation between the five factors; the lowest correlation was between CC and CSC, which was 0.397. Notably, the NC factor (anger expression) had high correlations with POS (*r* = 0.572), CC (*r* = 0.601), and CSC (*r* = 0.615). This is consistent with the previous research by Salguero et al. [18], which also found correlations between metacognitive beliefs and anger. In the end, correlation coefficients between the factors ranged between 0.397 and 0.727, indicating moderate and high correlations, and the correlations were significant. In addition, the total reliability of this research was 0.6 or higher, indicating that the MCRS was internally consistent.

This research has significance in that the reliability of the MCRS, a revised version of the MCQ that was supplemented by presenting the concepts of anger and anxiety, were ascertained with people with schizophrenia. Further, the study confirmed the high correlation between dysfunctional metacognitive beliefs and anger expression. Given the verification of the MCRS, follow-up research will be conducted to ensure that the reliability of the treatment, diagnosis, and rating of people with schizophrenia can be raised and also be used for future research.

## 5. Conclusions

This research was conducted to check the reliability of the MCRS, which revised and supplemented the MCQ to measure dysfunctional metacognition by adding the anger and anxiety concepts, in people with schizophrenia.

The MCRS consisted of five factors, with six items per factor. The MCRS was found to be reliable for measuring the metacognition of people with schizophrenia. The traditional standards of satisfactory reliability were met, but it is difficult to say that reliability was high. This research has significance as the first study to determine the metacognition of people with schizophrenia by adding anger and anxiety concepts.

There are several limitations to this study. Firstly, in this study we used EFA as an exploratory way of data analyses. Given the limitations associated with this method, future research should test replicability of the factorial structure using CFA methods. Secondly, although the reliability standards of the MCRS were generally met, they did not reach the optimal standards. The validity needs to be improved using the tools or reference instruments that confirm differences between the patient and control groups, and item correction may be required to improve internal consistency. Thirdly, this study targeted 184 people with schizophrenia hospitalized in mental hospitals, which limits the generalizability of the results. Future research should include people with schizophrenia under treatment and rehabilitation in various environments. Fourthly, the structure of the scales should be replicated in independent samples. Fifthly, mean comparisons with a healthy control group and possibly other patient samples should be provided. Sixthly, construct validity should be tested with other measurement instruments. Lastly, this study used a subjective measurement tool based on a self-reporting questionnaire. Thus, future research should develop and execute diverse measuring methods using objective third-party evaluation or physiological indicators.

## Figures and Tables

**Table 1 ijerph-18-06853-t001:** Demographic characteristics.

Variable	Item	Frequency	Ratio
Gender	Male	109	59.2
	Female	75	40.8
Level of education	Dropped out of middle school or earlier	99	53.8
	Graduated from middle school or received a higher level of education	85	46.2
Marital status	Unmarried	108	58.7
	Divorced	28	15.2
	Other	48	26.1
Age	49 or under	82	44.6
	50 or over	102	55.4
Residential type	Alone	55	29.9
	With family and friends	66	35.9
	Other	63	34.2
Status of taking prescriptions for mental health disease	Yes	176	95.7
	No	8	4.3
Variable	Frequency	Mean	SD
Age	184	50.70	12.51
Age of onset	177	30.17	13.01
No. of hospitalizations at a mental hospital	184	6.14	9.41

Note: *n* = 184.

**Table 2 ijerph-18-06853-t002:** Reliability of each factor.

Factor	POS	NEG	CC	NC (Anger Expression)	CSC
Loading of	Item loading	Item loading	Item loading	Item loading	Item loading
each item	MCRS 1/0.599	MCRS 2/0.766	MCRS 8/0.759	MCRS 6/0.781	MCRS 3/0.675
	MCRS 7/0.664	MCRS 4/0.762	MCRS 14/0.751	MCRS 13/0.639	MCRS 5/0.439
	MCRS 10/0.719	MCRS 9/0.833	MCRS 17/0.716	MCRS 20/0.782	MCRS 12/0.693
	MCRS 19/0.756	MCRS 11/0.743	MCRS 24/0.715	MCRS 22/0.757	MCRS 16/0.806
	MCRS 23/0.459	MCRS 15/0.810	MCRS 26/0.684	MCRS 25/0.498	MCRS 18/0.684
	MCRS 28/0.370	MCRS 21/0.764	MCRS 29/0.737	MCRS 27/0.278	MCRS 30/0.598
Eigenvalue	2.236	3.655	3.174	2.529	2.604
Cronbach’s α	0.643	0.871	0.821	0.696	0.731

Note: *n* = 184. POS: positive belief in worry, NEG: negative belief that worries are uncontrollable and dangerous, CC: cognitive confidence, NC: need for control, CSC: cognitive self-consciousness, MCRS: metacognitive rating scale.

**Table 3 ijerph-18-06853-t003:** Test for differences between the means by gender, level of education, and age.

Factor	Gender	Total (*n*)	Mean	SD	*t*	*p*	Cohen’s *d*	Level of Education	Total (*n*)	Mean	SD	*t*	*p*	Cohen’s *d*	Age	Total (*n*)	Mean	SD	*t*	*p*	Cohen’s *d*
POS	Male	109	16.36	4.82	−1.447	0.150	0.216	Incomplete middle school or lower	99	15.92	4.69	−2.651	0.009	0.393	≤ 49	82	16.77	4.97	−0.049	0.961	0.006
	Female	75	17.41	4.92				Middle school or above	85	17.80	4.92				≥ 50	102	16.80	4.82			
NEG	Male	109	15.07	6.42	−1.316	0.190	0.198	Incomplete middle school or lower	99	14.69	5.74	−2.061	0.041	0.308	≤ 49	82	16.20	6.80	1.181	0.239	0.176
	Female	75	16.32	6.17				Middle school or above	85	16.62	6.83				≥ 50	102	15.09	5.91			
CC	Male	109	14.05	6.30	0.668	0.505	0.097	Incomplete middle school or lower	99	13.34	5.61	−1.182	0.239	0.174	≤ 49	82	13.79	6.03	−0.047	0.963	0.006
	Female	75	13.48	5.16				Middle school or above	85	14.36	6.11				≥ 50	102	13.83	5.73			
NC (anger expression)	Male	109	15.38	5.57	0.239	0.811	0.036	Incomplete middle school or lower	99	14.71	4.78	−1.628	0.106	0.244	≤ 49	82	15.83	5.58	1.225	0.222	0.182
	Female	75	15.19	4.84				Middle school or above	85	15.99	5.75				≥ 50	102	14.87	5.00			
CSC	Male	109	17.07	5.27	−0.440	0.661	0.067	Incomplete middle school or lower	99	16.65	4.99	−1.571	0.118	0.231	≤ 49	82	17.72	5.74	1.144	0.254	0.170
	Female	75	17.43	5.47				Middle school or above	85	17.88	5.68				≥ 50	102	16.81	4.99			

Note: *n* = 184. POS: positive belief in worry, NEG: negative belief that worries are uncontrollable and dangerous, CC: cognitive confidence, NC: need for contrl, CSC: cognitive self-consciousness.

**Table 4 ijerph-18-06853-t004:** Test for differences between the means by the marital status and residential type.

Factor	Marital Status	Total (*n*)	Mean	SD	*t*	*p*	Cohen’s *d*	Residential Type	Total (*n*)	Mean	SD	*t*	*p*	Cohen’s *d*
POS	Unmarried	108	16.81	4.88	2.634	0.075	0.171	Alone	55	15.93	5.01	1.963	0.143	0.146
	Divorced	28	18.43	4.86				Family, friends	66	16.65	4.51			
	Other	48	15.79	4.72				Other	63	17.68	5.04			
NEG	Unmarried	108	15.35	6.52	1.522	0.221	0.129	Alone	55	15.64	6.67	0.585	0.558	0.080
	Divorced	28	17.46	5.75				Family, friends	66	14.97	5.77			
	Other	48	15.00	6.12				Other	63	16.17	6.62			
CC	Unmarried	108	13.81	6.30	1.405	0.248	0.124	Alone	55	15.00	6.20	1.638	0.197	0.134
	Divorced	28	15.29	5.56				Family, friends	66	13.21	5.23			
	Other	48	12.96	4.80				Other	63	13.41	6.09			
NC (anger expression)	Unmarried	108	15.59	5.41	2.834	0.061	0.176	Alone	55	15.40	6.05	0.121	0.886	0.036
	Divorced	28	16.61	5.04				Family, friend	66	15.05	5.09			
	Other	48	13.88	4.88				Other	63	15.48	4.80			
CSC	Unmarried	108	17.03	5.38	0.801	0.451	0.094	Alone	55	16.93	5.75	0.688	0.504	0.087
	Divorced	28	18.39	4.72				Family, friends	66	16.85	5.21			
	Other	48	16.96	5.61				Other	63	17.86	5.14			

Note: *n* = 184. POS: positive belief in worry, NEG: negative belief that worries are uncontrollable and dangerous, CC: cognitive confidence, NC: need for control, CSC: cognitive self-consciousness.

**Table 5 ijerph-18-06853-t005:** Correlation analysis of each factor.

Correlation Analysis	POS	NEG	CC	NC (Anger Expression)	CSC
POS	1				
NEG	0.607 **	1			
CC	0.434 **	0.590 **	1		
NC (anger expression)	0.572 **	0.727 **	0.601 **	1	
CSC	0.665 **	0.700 **	0.397 **	0.615 **	1

Note: *n* = 184. POS: positive belief in worry, NEG: negative belief that worries are uncontrollable and dangerous, CC: cognitive confidence, NC: need for control, CSC: cognitive self-consciousness; ** *p* < 0.01.

## Data Availability

The data presented in this study are available on request to the authors. Some variables are restricted to preserve the anonymity of study participants.

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
