# Peer review of "Metacognitive Rating Scale: A Study Applying a Korean Translation to Individuals with Schizophrenia"

_ijerph, 2021, doi:10.3390/ijerph18136853_

Round 1
Reviewer 1 Report
I believe that the authors carry out two studies in their research (tool validation and differences in the field of metacognitive beliefs). The authors should, however, first find out the factor structure (with all the requirements) of the adopted tool and subsequently suggest a model optimal for the relevant socio-cultural environment and age cohort (search for an acceptable model).
The authors do not mention the implementation of any preliminary research necessary in the case of transcultural research (see the protocol recommended within cross-cultural research – Klassen et al., 2009). Cross-cultural validation includes determining whether the tool, which originated in a particular socio-cultural context, is meaningfully applicable and, therefore, equivalent for use in another socio-cultural framework (Huang & Wong, 2014)
The authors should present an argument for the use of EFA instead of CFA (in the case of a model taken over and established from abroad, it is recommended to implement CFA). CFA verifies the latent factor structure of empirically obtained data with a preselected theory or assumption, which are frequently expressed by a structural model (Brown, 2006). EFA, conversely, is recommended when creating a new theory or research tool. Due to the nature of the data, I would recommend the MLR algorithm (Robust Maximum Likelihood) which has proven suitable properties for the calculation of ordinal data (Lei & Shieverdecker, 2020). At the same time, the following should be stated: variability of data and factor load for items of individual dimensions; acceptability of goodness-of-fit indices (CFI, TLI, RMSEA, … revision of items) and internal consistency; existence of a latent factor.
The authors state that by far not all dimensions have met the criteria of reliability and validity – how then can they identify differences (by gender, age, marital status, residential type)? It is methodologically wrong to detect differences in a situation where the psychometric properties of the tool are questionable (insufficiently acceptable).
Notes:
- The authors operate with metacognition as with a construct that is spread across the noncognitive and cognitive domain (metacognitive beliefs), although the metacognitive construct is usually separated from the noncognitive domain. Such an approach must be justified.
- The authors state that Cronbach’s α was 0.6 or higher, thus reliability was confirmed. The claim that 0.6 reliability is sufficient must be justified.
- Are the authors aware of studies that discuss the quality of tools (for example, in the context of correlation with performance; their construct validity) for determination of the level of metacognitive development based on the frequency of used strategies (Wirth & Leutner, 2008)? Already at the outset of the new millennium, the validity of this measurement procedure as well as the relevance of the use of the strategy highlighted in this way (number of used strategies and frequency of their use) was questioned (Artel, 2000; Pintrich et al., 2000; Neuenhaus, 2011) because the correlation of questionnaires measuring frequency of used strategies (“I have a poor memory”) with performance is negligible (Sperling et al., 2002; Cromley & Azevedo, 2006). Neuenhaus and her colleagues (2011) add that approaches determining the frequency of use of strategies actually measure whether a pupil recognized the actual strategy (rather than detecting metacognitive development). Regardless of the issue of domain-general x domain-specific method of measurement, most experts in educational psychology lean towards a domain-specific approach.
Artelt, C. (2000). Wie prädiktiv sind retrospektive Selbstberichte über den Gebrauch von Lernstrategien für strategisches Lernen? Zeitschrift für Pädagogische Psychologie, 14, 72‐84.
Brown, T. A. (2006). Confirmatory factor analysis for applied research. New York: Guilford.
Cromley, J. G., & Azevedo, R. (2006). Self‐report of reading comprehension strategies: What are we measuring? Metacognition and Learning, 1, 229‐247.
Huang, W. Y., & Wong, S. H. (2014). Cross-cultural validation. In A. C. Michalos (Ed.), Encyclopedia of quality of life and well-being research (pp. 1369–1371). Dordrecht: Springer.
Klassen, M. R., Bong, M., Usher, L. E., Chong, H. W., Huan, S. V., Wong, F. Y. I., & Georgiou, T., (2009). Exploring the validity of a teachers’ self-efficacy scale in five countries. Contemporary Educational Psychology, 34, 67-76.
Lei, P., & Shiverdecker, L. K. (2020). Performance of estmators for Confirmatory factor analysis of ordinal variables with missing data. Structural equation modeling: A Multidisciplinary Journal, 27(4), 584-601.
Neuenhaus, N. (2011). Metakognition und Leistung: Eine Längsschnittuntersuchung in den Bereichen Lesen und Englisch bei Schülerinnen und Schülern der fünften und sechsten Jahrgangsstufe (Doctoral dissertation, Universität Otto-Friedrich, Bamberg, Germany) [online]. [cit. 2021-02-02].
Neuenhaus, N., Artelt, C., Lingel, K., & Schneider, W. (2011). Fifth graders metacognitive knowledge: general or domain-specific? European Journal of Psychology of Education, 26(2), 163-178.
Pintrich, P. R. (2000). The role of goal orientation in self-regulated learning. In M. Boekarts & P. R. Pintrich (Eds.), Handbook of self-regulation (pp. 451-502). San Diego, CA, Academic Press.
Sperling, R. A., Howard, B. C., Miller, L. A., & Murphy, C. (2002). Measures of children's knowledge and regulation of cognition. Contemporary Educational Psychology, 27(1), 51-79.
Wirth, J., & Leutner, D. (2008). Self-regulated learning as a competence. Zeitschrift für Psychologie/Journal of Psychology, 216, 102-110.
Author Response
We thank the reviewers for their careful reading of the manuscript and their constructive remarks. We have addressed the comments, which enabled us to improve and clarify the manuscript. Please find below a detailed point-by-point response to all comments (reviewers’ comments in black, our replies in blue). Line numbering refers to the revised manuscript, which is attached as a supplement to the response to the editor.
Reviewer # 1
I believe that the authors carry out two studies in their research (tool validation and differences in the field of metacognitive beliefs). The authors should, however, first find out the factor structure (with all the requirements) of the adopted tool and subsequently suggest a model optimal for the relevant socio-cultural environment and age cohort (search for an acceptable model).
Reply:
This study examined the relationship between the Metacognitive Rating Scale (MCRS) and metacognitive belief sub-factors and general characteristics of schizophrenia patients, including the concepts of anger and anxiety. The five factors of the Metacognitions Questionnaire (MCQ) were applied as is; we did not analyze the factor structure.
The authors do not mention the implementation of any preliminary research necessary in the case of transcultural research (see the protocol recommended within cross-cultural research – Klassen et al., 2009). Cross-cultural validation includes determining whether the tool, which originated in a particular socio-cultural context, is meaningfully applicable and, therefore, equivalent for use in another socio-cultural framework (Huang & Wong, 2014)
Reply:
- According to Klassen et al. (2009), comparisons between countries in translational research requires confirmatory factor analysis (CFA). Therefore, This study has the limitation that it used EFA but not CFA.
- We reviewed Hunang & Wong (2014) to determine whether the tool was meaningful considering the sociocultural context. The process of developing this research tool reflected the guidelines for the process of cross-cultural adaptation of self-report measures (Beaton, Bombardier, Guillemin, & Ferraz, 2000); we added this reference to the manuscript.
The authors should present an argument for the use of EFA instead of CFA (in the case of a model taken over and established from abroad, it is recommended to implement CFA). CFA verifies the latent factor structure of empirically obtained data with a preselected theory or assumption, which are frequently expressed by a structural model (Brown, 2006).
Reply:
We reviewed Brown (2006). The metacognitive beliefs considered in this study are somewhat lacking theoretical background as the relationships with social psychological factors have not yet been clarified (Lenzo, Sardella, Martino, & Quattropani, 2020). Therefore, EFA was used to determine the structure of the data and for the development and validation analysis of the scales.
EFA, conversely, is recommended when creating a new theory or research tool. Due to the nature of the data, I would recommend the MLR algorithm (Robust Maximum Likelihood) which has proven suitable properties for the calculation of ordinal data (Lei & Shieverdecker, 2020). At the same time, the following should be stated: variability of data and factor load for items of individual dimensions; acceptability of goodness-of-fit indices (CFI, TLI, RMSEA, revision of items) and internal consistency; existence of a latent factor.
Reply:
The references presented by Lei & Shieverdecker (2020) were reviewed. Our study is an exploration of the relationship between factors and questions, without specific hypotheses; we concur that this is a limitation of this study.
The authors state that by far not all dimensions have met the criteria of reliability and validity – how then can they identify differences (by gender, age, marital status, residential type)? It is methodologically wrong to detect differences in a situation where the psychometric properties of the tool are questionable (insufficiently acceptable).
Reply
Most participants in this study did not graduate middle school, the average age was 50.7 years, and they had been hospitalized in psychiatric wards with schizophrenia for more than 20 years. There is little extant research on use of the MCQ with schizophrenia patients (Baumgartner et al., 2020).
Notes:
The authors operate with metacognition as with a construct that is spread across the noncognitive and cognitive domain (metacognitive beliefs), although the metacognitive construct is usually separated from the noncognitive domain. Such an approach must be justified.
Reply:
Metacognitive beliefs are not just metacognition or conceptions that are confined to cognitive domains. Metacognition includes self-awareness or the non-cognitive domains of an individual (Semeraro, Giofrè, Coppola, Lucangeli, & Cassibba, 2020)
The authors state that Cronbach’s α was 0.6 or higher, thus reliability was confirmed. The claim that 0.6 reliability is sufficient must be justified.
Reply:
Research shows (Kiliç, 2016) that values greater than .60 are acceptable. We added applicable references.
Are the authors aware of studies that discuss the quality of tools (for example, in the context of correlation with performance; their construct validity) for determination of the level of metacognitive development based on the frequency of used strategies (Wirth & Leutner, 2008)?
Reply:
We have read Wirth & Leutner (2008). The self-regulated executive function model (S-REF) model, which is the basis of this study, is that of Wells & Matthews (1994, 2008), which differs from the self-regulated learning presented in the reference(Wirth & Leutner (2008)).
Already at the outset of the new millennium, the validity of this measurement procedure as well as the relevance of the use of the strategy highlighted in this way (number of used strategies and frequency of their use) was questioned (Artel, 2000; Pintrich et al., 2000; Neuenhaus, 2011) because the correlation of questionnaires measuring frequency of used strategies (“I have a poor memory”) with performance is negligible (Sperling et al., 2002; Cromley & Azevedo, 2006).
Reply:
This study measured metacognitive beliefs of schizophrenia patients rather than metacognitive knowledge or academic achievement strategy of general students. Beliefs are an important non-cognitive aspect of the individual. Please consider the difference.
Neuenhaus and her colleagues (2011) add that approaches determining the frequency of use of strategies actually measure whether a pupil recognized the actual strategy (rather than detecting metacognitive development). Regardless of the issue of domain-general x domain-specific method of measurement, most experts in educational psychology lean towards a domain-specific approach.
Reply:
We reviewed the Neuenhaus et al. (2011) reference. The current study measured metacognitive beliefs in middle-aged schizophrenia patients, rather than metacognitive knowledge and metacognitive monitoring of students in the field of educational psychology. Our study explores how psychotic characteristics of schizophrenia appear in metacognitive belief measurements. Thank you.
Artelt, C. (2000). Wie prädiktiv sind retrospektive Selbstberichte über den Gebrauch von Lernstrategien für strategisches Lernen? Zeitschrift für Pädagogische Psychologie, 14, 72‐84.
Brown, T. A. (2006). Confirmatory factor analysis for applied research. New York: Guilford.
Cromley, J. G., & Azevedo, R. (2006). Self‐report of reading comprehension strategies: What are we measuring? Metacognition and Learning, 1, 229‐247.
Huang, W. Y., & Wong, S. H. (2014). Cross-cultural validation. In A. C. Michalos (Ed.), Encyclopedia of quality of life and well-being research (pp. 1369–1371). Dordrecht: Springer.
Klassen, M. R., Bong, M., Usher, L. E., Chong, H. W., Huan, S. V., Wong, F. Y. I., & Georgiou, T. (2009). Exploring the validity of a teachers’ self-efficacy scale in five countries. Contemporary Educational Psychology, 34, 67-76.
Lei, P., & Shiverdecker, L. K. (2020). Performance of estimators for confirmatory factor analysis of ordinal variables with missing data. Structural Equation Modeling, 27(4), 584-601.
Neuenhaus, N. (2011). Metakognition und Leistung: Eine Längsschnittuntersuchung in den Bereichen Lesen und Englisch bei Schülerinnen und Schülern der fünften und sechsten Jahrgangsstufe (Doctoral dissertation, Universität Otto-Friedrich, Bamberg, Germany).
Neuenhaus, N., Artelt, C., Lingel, K., & Schneider, W. (2011). Fifth graders’ metacognitive knowledge: General or domain-specific? European Journal of Psychology of Education, 26(2), 163-178.
Pintrich, P. R. (2000). The role of goal orientation in self-regulated learning. In M. Boekarts & P. R. Pintrich (Eds.), Handbook of self-regulation (pp. 451-502). San Diego, CA: Academic Press.
Sperling, R. A., Howard, B. C., Miller, L. A., & Murphy, C. (2002). Measures of children's knowledge and regulation of cognition. Contemporary Educational Psychology, 27(1), 51-79.
Wirth, J., & Leutner, D. (2008). Self-regulated learning as a competence. Zeitschrift für Psychologie/Journal of Psychology, 216, 102-110.
Baumgartner, J., Litvan, Z., Koch, M., Hinterbuchinger, B., Friedrich, F., Baumann, L., & Mossaheb, N. (2020). Metacognitive beliefs in individuals at risk for psychosis: a systematic review and meta-analysis of sex differences. neuropsychiatrie, 34(3), 108-115. doi:10.1007/s40211-020-00348-8
Beaton, D. E., Bombardier, C., Guillemin, F., & Ferraz, M. B. (2000). Guidelines for the process of cross-cultural adaptation of self-report measures. Spine (Phila Pa 1976), 25(24), 3186-3191. doi:10.1097/00007632-200012150-00014
Kiliç, S. (2016). Cronbach's alpha reliability coefficient. Psychiatry and Behavioral Sciences, 6(1), 47.
Lenzo, V., Sardella, A., Martino, G., & Quattropani, M. C. (2020). A Systematic Review of Metacognitive Beliefs in Chronic Medical Conditions. Frontiers In Psychology, 10(2875). doi:10.3389/fpsyg.2019.02875
Semeraro, C., Giofrè, D., Coppola, G., Lucangeli, D., & Cassibba, R. (2020). The role of cognitive and non-cognitive factors in mathematics achievement: The importance of the quality of the student-teacher relationship in middle school. PLoS ONE, 15(4), e0231381.

Reviewer 2 Report
The authors present results obtained from a Korean adaptation of the Metacognitive Rating Scale that was extended by two scales measuring anger and anxiety. The sample comprised N=184 adults diagnosed with schizophrenia. Results indicate generally satisfactory internal consistency and moderate to high intercorrelations of the scales. I think this is an informative contribution.
However, it could profit from some more efforts, in particular with respect to the psychometric analyses. Given that items were revised and added, their wordings should be made available, further, item statistics should be provided. The interpretation of the cumulative proportion of variance explained (across factors) as an indicator of validity seems unusual and should be removed. Corresponding interpretations in the Aims and Discussion sections should be revised accordingly.
Further, limitations such as a moderate sample size, the absence of a control group, and the absence of other instruments that would warrant inspection of construct validity should be clearly discussed. Most of my other remarks are rather minor and could be addressed easily, I think.
Introduction
- p. 2, para 2: "are high correlated with anger": highly?
- p. 2, para 3: "five factors that are distinguished conceptually, despite their correlations": Constructs can be correlated in line with theory. However, the positive relations of the meta-cognitive scales with anger and anxiety – that have been observed as well in previous research (e.g., Salguero et al., 2020) – call for an explanation. Possibly, this reflects better introspection in people with higher meta-cognitive abilities? Or could this be a confounding by self-report biases?
- p.2, para 6 (Aims of the Study): I suggest removing validity in the first aim (i.e. as indexed by cumulative proportion of variance explained).
Materials and Methods
- p. 2, last para: Some of the details were provided already in the Introduction. Possibly focus on the instrument that was actually employed in this study.
- p. 3, para 3: "Finally, the researchers added anger and anxiety concepts reflecting Korean cultural features in consideration of cultural differences." Given that also new scales were added, it would be interesting for the reader to add example items in the text or to make all items available in an appendix.
- p. 3: "conducted using SPSS Statistics 26.0": I think version 27 is the most current release. However, I do not think this would make much of a difference.
Results
- Table 1: "Age when schizophrenia occurred": Age when schizophrenia was diagnosed for the first time?
- p. 4, para 1: "exploratory factor analysis (EFA) was used in consideration of responses of people with schizophrenia whose level of education was low.": I am not sure if EFA is particularly suited to deal with responses from people with low educational level. Maybe just mention that EFA was used to infer the factors (principle components) that could potentially account for the observed correlations?
- p. 4, para 1: "the number of factors was when an eigenvalue was 1 or higher": The Kaiser-Guttman-Kriterium is a bit dated as it is affected by the number of items. The scree criterion (number of factors before the last major drop in Eigenvalues) may be more robust. (A parallel analysis is currently not offered in SPSS, as far as I know).
- p. 4, para 2: "if Cronbach's α was 0.6 or higher, reliability was confirmed": Following conventional criteria, this seems rather the lower bound for satisfactory reliability. However, most alphas are higher anyway. Further, the scales are short and the items may have been sufficiently heterogeneous to cover the full breadth of the construct. This could be discussed in a limitations section. Given that this is the first study to report psychometrics of this extended instrument, it would be informative to read item statistics (in particular, loadings).
- Table 2: I suggest removing the row "Cumulative %" as the factors seem to be re-ordered. The total proportion of variance accounted for by all factors could be given in the text.
- I suggest removing all statements referring to cumulative variance explanation as a criterion for the validity of a specific factor (e.g., "validity was only partially met as the total cumulative explanation rate was low at 37%"). If anything, cumulative variance explanation (i.e. after including the last factor) could be interpreted in the sense that the five factors jointly offer a fair account of the observed pattern of covariation of the items. This should be revised also in the Discussion.
- Table 3, 4, 5, 6, and 7: It could be informative to add a measure of effect size to this table (e.g., Cohen's d), as some of the differences between gender groups seem to be larger than others. Possibly low power to detect small differences given the moderate N could be discussed. Alternatively, the authors could suggest a minimum effect sizes that should be exceeded in order to interpret an effect as relevant. Possibly Tables 3-7 could be combined into one long table?
Discussion
- p. 8, para 3: "As each item's reliability was 0.684 or higher": Individual item reliabilities (e.g., factor loadings) are not reported in this manuscript.
- p. 9, para 1: "Given that the respondents' overall level of education was low ..., it would be impractical to conclude that a higher education level is related to higher meta-cognitive beliefs based on graduation from middle school.": Well, this is exactly what the significant difference says. Possibly it indicates that people with a higher educational degree (as a proxy for cognitive functioning) are better aware of their cognitive dys-/functioning?
- p. 9, para 2: "respondents' metacognition scores for each factor ... were generally higher than those ... found in a study by Wells and Cart-wright-Hatton": The interpretation offered by the authors is plausible. However, given that items were revised and adjusted to the Korean culture, caution should be exercised when comparing numeric values.
Conclusions
- p. 9, para 6: "they did not reach optimal standards; thus repeated research should be conducted": Repetition alone will not likely improve the psychometrics of the instrument; possibly item revision is at place to improve internal consistency. Congruent validity has to be demonstrated with established instruments or criterion validity by demonstrating relevant group differences (e.g., patients vs. controls).
- p. 9, para 7: "Thus, future research should develop and execute diverse measuring methods using objective third-party evaluation or physiological indicators.": Yes, this would be desirable, although the correlations of self-report scales with objective (laboratory) measures are usually weak to absent.
Author Response
We thank the reviewers for their careful reading of the manuscript and their constructive remarks. We have addressed the comments, which enabled us to improve and clarify the manuscript. Please find below a detailed point-by-point response to all comments (reviewers’ comments in black, our replies in blue). Line numbering refers to the revised manuscript, which is attached as a supplement to the response to the editor.
Reviewer # 2
The authors present results obtained from a Korean adaptation of the Metacognitive Rating Scale that was extended by two scales measuring anger and anxiety. The sample comprised N=184 adults diagnosed with schizophrenia. Results indicate generally satisfactory internal consistency and moderate to high inter-correlations of the scales. I think this is an informative contribution.
However, it could profit from some more efforts, in particular with respect to the psychometric analyses. Given that items were revised and added, their wordings should be made available, further, item statistics should be provided. The interpretation of the cumulative proportion of variance explained (across factors) as an indicator of validity seems unusual and should be removed. Corresponding interpretations in the Aims and Discussion sections should be revised accordingly.
Reply:
Thank you so much for your detailed remarks.
The factor analysis was exploratory, with the aim of helping inform the direction of future research in a field that is not yet systematized in theory. There are few extant studies of the metacognitive beliefs of schizophrenia patients. Since the relationship between metacognitive beliefs and psychosocial factors has not been clarified yet, it was not possible to present a clear hypothesis or conduct confirmatory factor analysis.
We have deleted the cumulative percentage of variance described in the validity and validity indicators section and deleted the applicable sections of the research purpose and discussion.
Further, limitations such as a moderate sample size, the absence of a control group, and the absence of other instruments that would warrant inspection of construct validity should be clearly discussed. Most of my other remarks are rather minor and could be addressed easily, I think.
Reply: We have added discussion of the limitations of this study, namely those pertaining to sample representativeness, the absence of a control group, and the absence of other tools that could have been used to verify construct validity.
Introduction
- p. 2, para 2: "are high correlated with anger": highly?
Reply:
“High” was changed to “highly.” Thank you for noting this typo.
- p. 2, para 3: "five factors that are distinguished conceptually, despite their correlations": Constructs can be correlated in line with theory. However, the positive relations of the meta-cognitive scales with anger and anxiety – that have been observed as well in previous research (e.g., Salguero et al., 2020) – call for an explanation. Possibly, this reflects better introspection in people with higher meta-cognitive abilities? Or could this be a confounding by self-report biases?
Reply: Thank you. It has been added as follows.
This does not indicate that better self-reflection was present in people with high metacognitive abilities.
- p.2, para 6 (Aims of the Study): I suggest removing validity in the first aim (i.e. as indexed by cumulative proportion of variance explained).
Reply:
Thank you for the suggestion. It has been modified.
Materials and Methods
- p. 2, last para: Some of the details were provided already in the Introduction. Possibly focus on the instrument that was actually employed in this study.
Reply:
We have supplemented the content with focus on the MCRS.
- p. 3, para 3: "Finally, the researchers added anger and anxiety concepts reflecting Korean cultural features in consideration of cultural differences." Given that also new scales were added, it would be interesting for the reader to add example items in the text or to make all items available in an appendix.
Reply:
We now list the scale with the added concepts of anger and anxiety as an appendix.
- p. 3: "conducted using SPSS Statistics 26.0": I think version 27 is the most current release. However, I do not think this would make much of a difference.
Reply:
We agree, thank you.
Results
- Table 1: "Age when schizophrenia occurred": Age when schizophrenia was diagnosed for the first time?
Reply:
Yes, we refer to age of onset; the text has been changed to clarify this.
- p. 4, para 1: "exploratory factor analysis (EFA) was used in consideration of responses of people with schizophrenia whose level of education was low.": I am not sure if EFA is particularly suited to deal with responses from people with low educational level. Maybe just mention that EFA was used to infer the factors (principle components) that could potentially account for the observed correlations?
Reply:
Thank you for this suggestion. It has been corrected; EFA was used to infer the factors (principle components) that could potentially account for the observed correlation.
- p. 4, para 1: " The number of factors was when an eigenvalue was 1 or higher": The Kaiser-Guttman-Kriterium is a bit dated as it is affected by the number of items. The scree criterion (number of factors before the last major drop in Eigenvalues) may be more robust. (A parallel analysis is currently not offered in SPSS, as far as I know).
Reply:
Thank you for pointing this out; Parallel analysis is not provided to SPSS as agreed.
- p. 4, para 2: "if Cronbach's α was 0.6 or higher, reliability was confirmed": Following conventional criteria, this seems rather the lower bound for satisfactory reliability. However, most alphas are higher anyway. Further, the scales are short and the items may have been sufficiently heterogeneous to cover the full breadth of the construct. This could be discussed in a limitations section. Given that this is the first study to report psychometrics of this extended instrument, it would be informative to read item statistics (in particular, loadings).
Reply:
This is the first study to use the MCRS; reliability was confirmed according to traditional standards, but it is difficult to say that reliability was high.
- Table 2: I suggest removing the row "Cumulative %" as the factors seem to be re-ordered. The total proportion of variance accounted for by all factors could be given in the text.
Reply:
We have removed the validity and cumulative % from Table 2, the text, and the discussion. Thank you.
- I suggest removing all statements referring to cumulative variance explanation as a criterion for the validity of a specific factor (e.g., "validity was only partially met as the total cumulative explanation rate was low at 37%"). If anything, cumulative variance explanation (i.e. after including the last factor) could be interpreted in the sense that the five factors jointly offer a fair account of the observed pattern of covariation of the items. This should be revised also in the Discussion.
- Table 3, 4, 5, 6, and 7: It could be informative to add a measure of effect size to this table (e.g., Cohen's d), as some of the differences between gender groups seem to be larger than others. Possibly low power to detect small differences given the moderate N could be discussed. Alternatively, the authors could suggest a minimum effect sizes that should be exceeded in order to interpret an effect as relevant. Possibly Tables 3-7 could be combined into one long table?
Reply:
The effect sizes have been added to the table. Tables 3 to 7 have been combined.
Discussion
- p. 8, para 3: "As each item's reliability was 0.684 or higher": Individual item reliabilities (e.g., factor loadings) are not reported in this manuscript.
Reply:
In Table 2, we presented individual item reliabilities (e.g., factor loadings).
- p. 9, para 1: "Given that the respondents' overall level of education was low ..., it would be impractical to conclude that a higher education level is related to higher meta-cognitive beliefs based on graduation from middle school.": Well, this is exactly what the significant difference says. Possibly it indicates that people with a higher educational degree (as a proxy for cognitive functioning) are better aware of their cognitive dys-/functioning?
Reply:
Yes; people with higher education levels are more aware of their cognitive function or disability, so they reduce that behavior. That is, as you explain, people who are better aware of cognitive impairment tend to have a higher level of education.
- p. 9, para 2: "respondents' metacognition scores for each factor ... were generally higher than those ... found in a study by Wells and Cart-wright-Hatton": The interpretation offered by the authors is plausible. However, given that items were revised and adjusted to the Korean culture, caution should be exercised when comparing numeric values.
Reply:
We revised the text to recognize that the MCRS needs to be carefully compared considering that the original tools were revised and supplemented considering cultural differences. That is, dysfunctional metacognitive beliefs of schizophrenia patients were delineated, but it is unreasonable to simply compare the results of the MCRS and the original tool, given the culture-specific modifications we applied.
Conclusions
- p. 9, para 6: "they did not reach optimal standards; thus repeated research should be conducted": Repetition alone will not likely improve the psychometrics of the instrument; possibly item revision is at place to improve internal consistency. Congruent validity has to be demonstrated with established instruments or criterion validity by demonstrating relevant group differences (e.g., patients vs. controls).
Reply:
The text was modified to emphasize that it is necessary to improve validity using tools or reference instruments that confirm differences between patient and control groups, and that item correction may be required to improve internal consistency.
- p. 9, para 7: "Thus, future research should develop and execute diverse measuring methods using objective third-party evaluation or physiological indicators.": Yes, this would be desirable, although the correlations of self-report scales with objective (laboratory) measures are usually weak to absent.
Reply:
Thank you.
Appendix
Metacognitive Rating Scale
MCRS
Modified MCQ(Cartwright-Hatton & Wells, 1997; Wells & Cartwright-Hatton, 2004)
This questionnaire is concerned with beliefs people have about their thinking. Listed below are a number of beliefs that people have expressed. Please read each item and say how much you generally agree with it by circling the appropriate number. Please respond to all the items, there are no right or wrong answers.
Gender: Age:
1 Strongly disagree; 2 Disagree; 3 Unsure; 4 Agree; 5 Strongly agree.
|
1. Worrying helps me to avoid problems in the future |
1 |
2 |
3 |
4 |
5 |
|
2. My worrying is dangerous for me |
1 |
2 |
3 |
4 |
5 |
|
3. I think a lot about my thoughts |
1 |
2 |
3 |
4 |
5 |
|
4. When you become addicted to worry, even the slightest amount of stress lead to anxiety |
1 |
2 |
3 |
4 |
5 |
|
5. I am aware of the way my mind works when I am thinking through a problem |
1 |
2 |
3 |
4 |
5 |
|
6. I easily get angry when I feel irritated |
1 |
2 |
3 |
4 |
5 |
|
7. I need to worry in order to work well |
1 |
2 |
3 |
4 |
5 |
|
8. I have little confidence in my memory for words and names |
1 |
2 |
3 |
4 |
5 |
|
9. My worrying thoughts persist, no matter how I try to stop them |
1 |
2 |
3 |
4 |
5 |
|
10. Worrying helps me to avoid problems in the future |
1 |
2 |
3 |
4 |
5 |
|
11. I cannot ignore my worrying thoughts |
1 |
2 |
3 |
4 |
5 |
|
12. I monitor my thoughts |
1 |
2 |
3 |
4 |
5 |
|
13. I always try to suppress my anger |
1 |
2 |
3 |
4 |
5 |
|
14. My memory can mislead me at times |
1 |
2 |
3 |
4 |
5 |
|
15. My worrying could make me go mad |
1 |
2 |
3 |
4 |
5 |
|
16. I am constantly aware of my thought |
1 |
2 |
3 |
4 |
5 |
|
17. I have a poor memory |
1 |
2 |
3 |
4 |
5 |
|
18. I pay close attention to the way my mind works |
1 |
2 |
3 |
4 |
5 |
|
19. I need to worry in order to remain organized |
1 |
2 |
3 |
4 |
5 |
|
20. It is my weakness that I cannot control my anger. |
1 |
2 |
3 |
4 |
5 |
|
21. When I start worrying, I cannot stop |
1 |
2 |
3 |
4 |
5 |
|
22. I cannot control expressions of anger |
1 |
2 |
3 |
4 |
5 |
|
23. Worrying helps me to get things sorted out in my mind |
1 |
2 |
3 |
4 |
5 |
|
24. I have little confidence in my memory for places |
1 |
2 |
3 |
4 |
5 |
|
25. Expressing anger is bad |
1 |
2 |
3 |
4 |
5 |
|
26. I do not trust my memory |
1 |
2 |
3 |
4 |
5 |
|
27. I can express anger in words those are not aggressive |
1 |
2 |
3 |
4 |
5 |
|
28. Worrying helps me cope |
1 |
2 |
3 |
4 |
5 |
|
29. I have little confidence in my memory for actions |
1 |
2 |
3 |
4 |
5 |
|
30. I constantly examine my thoughts |
1 |
2 |
3 |
4 |
5 |
Please ensure that you have responded to all of the items - Thank You.
MCRS-SCORING KEY
Enter the number given by the subject for each item in the relevant box below and then sum the scores to produce a subscale total.
|
Positive beliefs about worry (POS) |
Negative beliefs about uncontrollability and danger of worry (NEG) |
Cognitive confidence (CC)
|
Need for control (NC)
|
Cognitive self-consciousness (CSC) |
|||||
|
1 |
|
2 |
|
8 |
|
6 |
|
3 |
|
|
7 |
|
4 |
|
14 |
|
13 |
|
5 |
|
|
10 |
|
9 |
|
17 |
|
20 |
|
12 |
|
|
19 |
|
11 |
|
24 |
|
22 |
|
16 |
|
|
23 |
|
15 |
|
26 |
|
25 |
|
18 |
|
|
28 |
|
21 |
|
29 |
|
27 |
|
30 |
|
|
total |
|
|
|
|
|
|
|
|
|
An overall total MCQ score can be obtained by summing the subscale totals.

Round 2
Reviewer 1 Report
I accept most of the authors´ reflections. I thank them (in general) for adequate and relevant comments, including references. Justification of authors´ approach is relevant. I have just one last issue:
From a methodological point of view, in my opinion, it is not possible to apply the MCQ tool in the context of correlation with another tool (MCRS) unless the psychometric properties of the tool (tools) are known from preliminary study (tools applied to a similar sample before the main study; we must/should know factor structure, factor loadings, reliability…goodness-of-fit indices). Can the authors demonstrate (references) that their approach is justifiable in the above mentioned context?
Author Response
Thank you for your valuable comments.
Exploratory factor analysis is a method of extracting factors through results without a theoretical background. Hence, this is why we agree with you that it is necessary to implement a preliminary study.
However, this study did not present a preliminary study and was bound to the factors of the existing MCQ because of the following reasons.
First, exploratory factor analysis binds factors based on the respondents’ results; however, the respondents of this study were schizophrenia patients. The factors were set based on the results of previous studies because various variables such as the characteristics of the disease, age, gender, income level, and education level of the respondents can affect the results. Because the respondents’ characteristics affect the results, even if the results are different from those of the previous studies, the survey cannot be said to be wrong; however, we explain that the factors of the previous studies were used considering the limitations of exploratory factor analysis (Stapleton, 1997).
Second, if the factors are tied up only by the respondents’ response results without including the factors of the existing MCQ, the factors that are not related can be combined and the interpretation of the factors can become awkward or complicated (Nunnally, 1978). We think anyone who has conducted exploratory factor analysis has experienced it once. Because factor analysis is analyzed by correlation between variables, it can be combined as one factor if it is a variable with a statistically high correlation.
Third, the purpose of the preliminary study in exploratory factor analysis is to identify the exact factors (Fabrigar, Wegener, MacCallum, & Strahan, 1999). We apply the factors of the existing MCQ considering the problem of exploratory factor analysis described above.
References
Stapleton, C. D. (1997). Basic Concepts and Procedures of Confirmatory Factor Analysis.
Nunnally J.C. (1978) An Overview of Psychological Measurement. In: Wolman B.B. (eds) Clinical Diagnosis of Mental Disorders. Springer, Boston, MA. https://doi.org/10.1007/978-1-4684-2490-4_4
Fabrigar, L. R., Wegener, D. T., MacCallum, R. C., & Strahan, E. J. (1999). Evaluating the use of exploratory factor analysis in psychological research. Psychological Methods, 4(3), 272–299. https://doi.org/10.1037/1082-989X.4.3.272

Reviewer 2 Report
The authors have adequately addressed most concerns raised in my previous review. Only few issues remain that should be considered (see below).
I agree with R1 that the way of data analysis is arguably traditional. Future studies in this line of research should use state of the art methods, including CFA and test of measurement invariance.
- Table 2: I suggest changing "Item/Reliability" into "Item Loading". These could be interpreted both as reliability and validity.
- Conclusion: I suggest rewording the sentence: "But reliability was confirmed according to traditional standards, but it is difficult to say that reliability was high." Maybe, "Traditional standards of satisfactory reliability were met."
- I agree with R1 that CFA methods are better suited as a test of theory. However, I do not think the following sentence reads particularly well. "First, we only have access to SPSS and not the addon structural equation modeling program AMOS, so we could not conduct a CFA. But in that case, we acknowledge a limitation that CFA was not performed and suggest it be used for future research." I would suggest something like "In this study we used EFA as an exploratory way of data analyses. Given the limitations associated with this method, future research should test replicability of the factorial structure using CFA methods."
- Further, the limitations section should possibly comprise suggestions for future studies: (1) The structure of the scales should be replicated in independent samples. (2) Mean comparisons with a healthy control group and possibly other patient samples should be provided. (3) Construct validity should be tested with other measurement instruments. As also highlighted by R1, the virtual absence of meaningful relations between objective and self-report measures should be discussed in this context.
- Appendix: The table with item wordings is informative. However, the response scales can be removed to save space. (Just mention that five-point Likert scales were used.) A column indicating the scale to which the item belongs would be informative. Then, the table with scoring keys could be removed.
Author Response
Thank you so much for your comments.
I agree with R1 that the way of data analysis is arguably traditional. Future studies in this line of research should use state of the art methods, including CFA and test of measurement invariance.
We also agree. Thank you.
- Table 2: I suggest changing "Item/Reliability" into "Item Loading". These could be interpreted both as reliability and validity.
Reply:
Thank you. “Item/Reliability” was changed to “Item Loading.”
- Conclusion: I suggest rewording the sentence: "But reliability was confirmed according to traditional standards, but it is difficult to say that reliability was high." Maybe, "Traditional standards of satisfactory reliability were met."
Reply:
Thank you for the suggestion. This has been modified as follows.
Traditional standards of satisfactory reliability were met, but it is difficult to say that reliability was high.
- I agree with R1 that CFA methods are better suited as a test of theory. However, I do not think the following sentence reads particularly well. "First, we only have access to SPSS and not the addon structural equation modeling program AMOS, so we could not conduct a CFA. But in that case, we acknowledge a limitation that CFA was not performed and suggest it be used for future research." I would suggest something like "In this study we used EFA as an exploratory way of data analyses. Given the limitations associated with this method, future research should test replicability of the factorial structure using CFA methods."
Reply:
We agree; the text has been changed to clarify this.
First, in this study, we used EFA as an exploratory way of data analyses. Given the limitations associated with this method, future research should test the replicability of the factorial structure using CFA methods.
- Further, the limitations section should possibly comprise suggestions for future studies: (1) The structure of the scales should be replicated in independent samples. (2) Mean comparisons with a healthy control group and possibly other patient samples should be provided. (3) Construct validity should be tested with other measurement instruments. As also highlighted by R1, the virtual absence of meaningful relations between objective and self-report measures should be discussed in this context.
Reply:
Thank you. The following text has been added.
Fourth, the structure of the scales should be replicated in independent samples. Fifth, mean comparisons with a healthy control group and possibly other patient samples should be provided. Sixth, construct validity should be tested with other measurement instruments.
- Appendix: The table with item wordings is informative. However, the response scales can be removed to save space. (Just mention that five-point Likert scales were used.) A column indicating the scale to which the item belongs would be informative. Then, the table with scoring keys could be removed.
Reply:
Thank you for the suggestion. This has been modified.
